

# Surgical outcomes in patients with Achilles tendon rupture—a retrospective study

Hüseyin Kürüm[1], Hacı Bayram Tosun[1], Faruk Aydemir[2], Orhan Ayas[1], Kübra Orhan Kürüm[3] and Funda İpekten[4]

[1] Orthopaedics and Traumatology, Elazığ Fethi Sekin Training and Research Hospital, Elazığ, Turkey
[2] Department of Emergency, Elazığ Fethi Sekin Training and Research Hospital, Elazığ, Turkey
[3] Physical Therapy and Rehabilitation, İnönü University Turgut Özal Medical Center, Malatya, Turkey
[4] Biostatistics, Faculty of Medicine, Adiyaman University, Adıyaman, Turkey

Corresponding author
Hüseyin Kürüm,
dr.hsynkrm@gmail.com

## ABSTRACT

**Background:** There are two main methods used to treat Achilles tendon rupture (ATR): conservative treatment and surgical intervention. Surgical techniques are divided into three main categories: open surgical repair, mini-open surgical repair, and percutaneous repair (PR). We aimed to compare clinical outcomes in individuals with ATR who were treated with PR, primary repair, and flexor hallucis longus augmentation (FHL-A) with those treated with V-Y plasty and FHL-A.

**Methods:** The study involved 54 patients who underwent ATR surgical intervention retrospectively. Thirty-two of these were identified as acute and 22 were chronic rupture patients. PR was performed in 32 patients, primary repair and FHL-A in 14 patients, and V-Y plasty and FHL-A in eight patients.

**Results:** The mean forward jump was $142.69 \pm 7.14$ cm in individuals who received PR, $137.71 \pm 4.51$ cm in those who received primary repair + FHL-A, and $123.88 \pm 3.09$ cm in those who received V-Y plasty + FHL-A ($p < 0.001$). The decrease in the mean vertical jump distance on the operated side compared to the contralateral extremity was $0.97 \pm 0.93$, $2.07 \pm 0.99$, and $3.00 \pm 1.69$ cm in individuals who underwent PR, primary repair + FHL-A, and V-Y plasty + FHL-A, respectively ($p < 0.001$). The decrease in the mean dorsiflexion of the operated side ankle compared to the contralateral extremity was found to be $4.34 \pm 1.18$, $1.93 \pm 1.07$, and $2.38 \pm 0.92$ degrees in individuals who underwent PR, primary repair + FHL-A, and V-Y plasty + FHL-A, respectively ($p < 0.001$).

**Conclusion:** Although no surgical technique is completely superior to another, better performance tests were observed after PR repair compared to open surgery in individuals involved in sports, but rerupture, dorsiflexion restriction, and painful ankle were disadvantages. FHL-A, which has gained popularity in recent years, showed better performance in tests by contributing to more stable and stronger ankles in sporting individuals with chronic ruptures who had undergone open surgery. FHL-A can be utilized in addition to primary surgical intervention in individuals with high performance expectations before and after injury.

## INTRODUCTION

The Achilles tendon, with a length of approximately 12–15 cm, is the strongest and thickest tendon in the human body. Achilles tendon rupture (ATR) constitutes 20% of all tendon ruptures (*Gillies & Chalmers, 1970*).

There are two main methods of treating ATR: conservative treatment and surgical intervention (*Liu et al., 2018*; *Lee, Martin & Thelen, 2017*). Surgical techniques are divided into three main categories: open surgical repair, mini-open surgical repair, and percutaneous repair (PR) (*Hsu et al., 2015*).

Despite the superior wound site outcomes associated with the PR technique, which were initially described by *Ma & Griffith (1977)* in 1977, the technique has its own set of disadvantages. These include a reduction in the initial force provided by approximately 50%, an elevated risk for sural nerve injury, and higher rerupture rates compared to open repair (*Hockenbury & Johns, 1990*; *Bradley & Tibone, 1990*; *Jiang et al., 2021*). The Krackow suture technique (*Krackow, Thomas & Jones, 1988*) is the most commonly employed method in primary open repair. Despite demonstrating favourable clinical outcomes, this technique has been linked to elevated infection rates, necessitating subsequent revision (*Khan & Carey Smith, 2010*; *McMahon, Smith & Hing, 2011*; *Nilsson-Helander et al., 2010*).

Chronic ATR refers to a tear in the Achilles tendon that has not been diagnosed or treated for 6 weeks or more (*Maffulli, 1999*). Treating chronic ATR is more technically demanding than repairing acute ruptures due to the retraction of the tendon ends and increasing deterioration of the surrounding soft tissues, which complicates the main repair process (*Maffulli, Via & Oliva, 2015*).

Several techniques have been identified to repair and rehabilitate chronic ATR, such as V-Y plasty, gastrocnemius fascial turndown flap, tendon transfers, allograft reconstruction, autograft reconstruction, synthetic graft augmentation, and biological matrix augmentation (*de Cesar Netto et al., 2017*; *Usuelli et al., 2017*). In V-Y plasty, the tendon ends are excised, and a V-shaped incision is made in the tendon aponeurosis, which thereby enables the tendon to be lengthened and reattached to the calcaneus. A number of surgeons employ the V-Y advancement technique in conjunction with another augmentation procedure, such as flexor hallucis longus augmentation (FHL-A) (*Kissel, Blacklidge & Crowley, 1994*; *Ponnapula & Aaranson, 2010*; *Elias et al., 2007*). The use of FHL-A has gained popularity in recent studies due to its effectiveness in providing reconstruction and augmentation in chronic ATR. FHL-A is readily accessible due to its proximity to the Achilles tendon, has minimal donor site morbidity, and is a surgical augmentation method with low rates of neurovascular injury and wound site complications (*Oksanen et al., 2014*; *Maffulli, Aicale & Tarantino, 2017*).

The high rates of rerupture after conservative treatment increase the tendency towards surgical intervention in young, active individuals (*Jiang et al., 2012*; *Wilkins & Bisson, 2012*; *Soroceanu et al., 2012*). Currently, there is no gold standard for the optimal management of ATR or preferred method of surgical intervention. This study aimed to contribute to the literature by identifying the most appropriate surgical method after

Achilles rupture in active individuals with sports activities. To our knowledge, this is the first study to compare the functional outcomes and possible complications of PR, open surgical repair + FHL-A, and open V-Y plasty + FHL-A. This study evaluated patients who underwent three types of surgical intervention for ATR across several parameters, including infection rates, nerve damage, wound complications, time to return to work and/or sports activities, clinical outcomes assessed using the American Orthopedic Foot and Ankle Society (AOFAS) scoring system and Victoria Institute of Sport Assessment-Achilles (VISA-A) scale, balance capabilities, ankle plantar flexion, dorsiflexion angles, heel rise performances, and jumping performances.

## METHODS

In this study, 54 patients who underwent ATR surgical intervention between 2017 and 2023 were retrospectively evaluated. Thirty-two of these were identified as acute and 22 were chronic rupture patients. Patients who presented at our hospital less than 6 weeks post-injury were classified as having acute ruptures, whereas those who presented after 6 weeks were classified as having chronic ruptures. All 32 patients who presented acutely underwent PR, the 14 patients with neglected Achilles rupture and a gap size of less than 3 cm underwent primary repair + FHL-A, and the eight patients with a neglected gap size of more than 3 cm underwent V-Y plasty + FHL-A. The inclusion criteria included individuals between the ages of 18–54 years who participated in recreational sports activities before the rupture or who practiced sports at least once a week and were amateur or elite athletes, had ruptures within a distance of 2–10 cm from the Achilles insertion site, were treated with surgical repair, performed daily activities not limited before injury, and had preoperative ultrasound or MRI describing the site of rupture. The study protocol was approved by the Fırat University Hospital Human Subject Research Ethics Committee (2023/13–38). Data collection and evaluation were conducted in accordance with the Helsinki Declaration. All patients were informed about the treatment, and written consent was obtained.

A total of three patients had a history of diabetes mellitus, while two had a history of steroid injections resulting from Achilles tendinopathy. The presence of these comorbid conditions did not yield statistically significant differences between the groups. Among those who received PR, three were elite athletes, seven were amateur athletes, and 22 had a rupture while performing their daily activities and sportive activity. The patients undergoing primary repair + FHL-A were not amateur or professional athletes. Of the 14 injuries, 12 occurred during sports activities and two occurred during daily activities. Of the individuals that received V-Y plasty and FHL-A, two ruptures had occurred during sportive activity and six during daily activities (Table 1). The surgical interventions were performed by two different orthopedic surgeons at two different hospitals. Subsequently, a postoperative evaluation was carried out by a physiotherapist and an orthopedic surgeon.

### Exclusion criteria

The study excluded patients who had undergone revision Achilles surgical intervention, who had previously received conservative treatment, with incomplete medical records,

**Table 1 Patients' demographic characteristics.**

| VARIABLES | | TOTAL (n = 54) | |
| --- | --- | --- | --- |
| AGE (year) | | mean ± SD | 34.70 ± 8.82 |
| GENDER | Male | n (%) | 42 (77.8) |
| | Female | | 12 (22.2) |
| SIDE | Right | n (%) | 39 (70.4) |
| | Left | | 16 (29.6) |
| INJURY TYPE | Acut repair (PR) | n (%) | 32 (59) |
| | Chronic repair (Primary repair + FHL-A and V-Y plasty + FHL-A) | | 22 (41) |
| SURGERY TYPE | PR | | 32 (59) |
| | Primary repair + FHL-A | | 14 (26) |
| | V-Y plasty + FHL-A | | 8 (15) |
| PR | Elite athletes | n (%) | 3 (5.5) |
| | Amateur athletes | | 7 (12.9) |
| | Non-athletes | | 22 (40.7) |
| ACTİVİTY | Sportive activity | n (%) | 37 (68.5) |
| | Daily activity | | 17 (31.5) |
| COMORBIDITY | Diabetes mellutus | n (%) | 2 (3.7) |
| | Steroid injection | | 3 (5.6) |
| | Absent | | 49 (90.7) |

**Note:**
 Data are expressed as mean ± standard deviation (SD) and number (n) (%).

with open injuries, who had undergone only FHL transfer due to chronic ATR or Achilles tendinopathy, who experienced injury in the opposite lower leg 6 months after the procedure, and who had a follow-up period of less than 1 year. The flowchart of the patient screening process is presented in Fig. 1.

## Diagnosis of ATR

All patients reported the following primary complaints: inability to raise their heel, weakness when climbing stairs, weariness, and pain. Calcaneal gait was also observed in all patients. The Thompson test was negative in eight patients (14.8%) and positive in 46 patients (85.2%). Patients with a positive Thompson test exhibited hyperdorsiflexion compared to the contralateral extremity. Following the physical examination, ultrasound and/or magnetic resonance imaging were used to confirm the diagnosis.

## Measurement of ankle range of motion

A goniometer was placed laterally to the joint with the patient in a prone position and the knee bent at a 90-degree angle. Subsequently, the range of motion (ROM) for ankle dorsiflexion and plantar flexion were measured and recorded.

## Forward and vertical jump performance

A total of 6 months after surgical intervention, patients underwent vertical and horizontal jump tests to assess their Achilles tendon strength. The forward jump test required

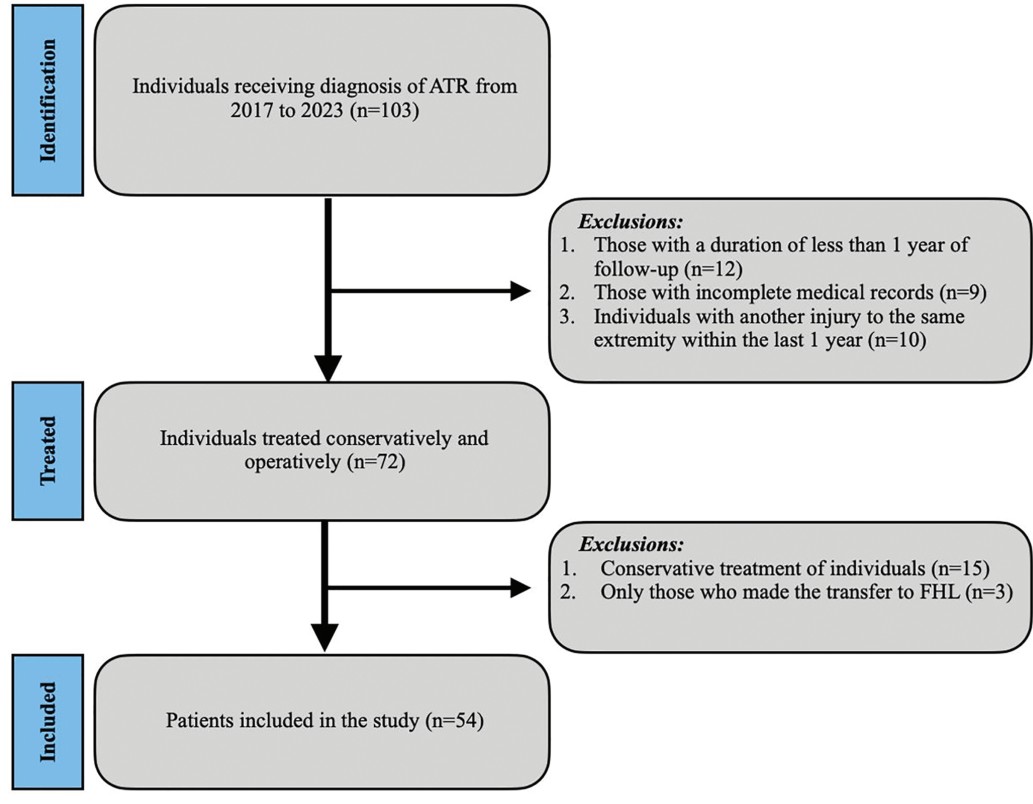

**Figure 1 Flowchart of individuals undergoing surgical treatment for ATR.**

participants to balance on one foot while keeping their toes positioned behind a designated line. Participants were then asked to jump as far as possible with a controlled landing. The test was conducted twice, and the best distance was measured in centimetres. The test commenced by assessing the non-operated extremity before proceeding to evaluate the operative one. During the vertical leap test, participants were instructed to achieve the maximum height possible while positioned in front of a wall. Prior to the examination, the patient's baseline arm length was measured in front of the testing platform. The vertical jump distance, measured in centimeters, was determined by calculating the difference between the jump distance and arm length (Fig. 2).

## Single leg stance test

The duration of patients' unipedal stance was recorded to determine their balance. Individuals were considered to have balance problems if they were unable to maintain a stable position on one leg for less than 10 s without making any corrective movements (Fig. 2).

## Heel rise test

The endurance of the plantar flexor muscles of the ankle was evaluated using the heel rise test. Individuals were permitted to touch anything with their fingers to establish their balance. In this position, they were instructed to elevate their heels in response to each
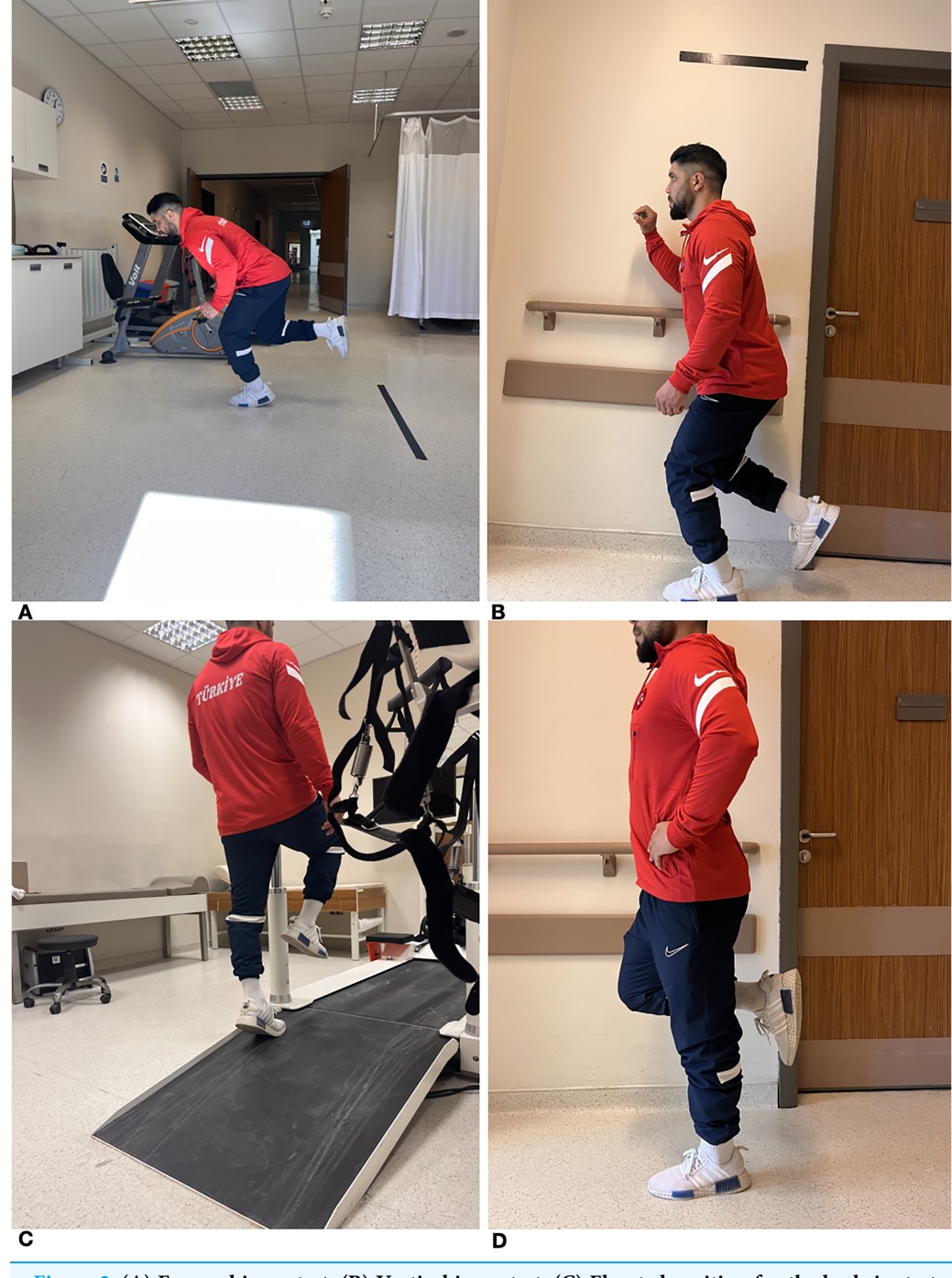

**Figure 2** (A) Forward jump test. (B) Vertical jump test. (C) Elevated position for the heel rise test. (D) Single Leg Stance test.

auditory signal, synchronized with a metronome set at a tempo of 30 beats per minute. Heel rises of approximately 5 cm were recorded. The test was demonstrated and explained to the patients by a physiotherapist. The unaffected side was tested first followed by the operated side (Fig. 2).

## Rehabilitation

Patients were called for examination on the 15th postoperative day, as well as at the first, second, third, 6th, and 12th months. Individuals who received PR were immobilized in a short leg circular cast with their leg in a plantarflexion for a duration of 2 weeks. On the 15th day, a short leg cast in a neutral position was applied to the patients at the outpatient clinic for control purposes. They were allowed to move with slight weight-bearing on tiptoes. At the conclusion of the initial month, the subjects were fitted with a controlled ankle movement (CAM) Achilles boot that could be adjusted to accommodate varying angles. They wore this boot for a duration of 1 month and mobilized bearing as much weight as tolerated. At the end of the second month, the patients were instructed to bear full weight and beginning ankle exercises. Patients who underwent open surgical repair were immobilized with a short leg circular cast with their leg in plantarflexion for a duration of 2 weeks and were allowed to move with minimal weight-bearing on their fingertips. On the 15th day of outpatient follow-up, a short leg circular cast was applied in the neutral position. At the end of the sixth week, the short leg circular cast was removed. Subsequently a CAM with adjustable angles was employed for a duration of 1 month, during which the individuals were encouraged to engage in walking with the maximum amount of weight they could endure. After a period of 2.5 months, the patient began to bear their entire weight on the affected limb, and gradually evolved to performing activities that involved ankle exercises. Every patient using the CAM Achilles boot commenced with active-passive motions. In the initial 6-month period following injury, low-impact activities were permitted, and subsequently, high-impact activities, including soccer and basketball, were allowed (*Willits et al., 2010*) (Table 2).

For the purposes of clinical assessment, the VISA-A scale at 6 months and AOFAS scoring system at 1 year were employed. The VISA-A scale was comprised of four categories: excellent (100–90 points), good (89–80 points), fair (79–70 points), and poor (below 70 points). The AOFAS scoring system was comprised of four categories: excellent (90–100 points), good (80–89 points), fair (70–79 points), and poor (below 70 points).

## Surgical technique

All procedures were performed under spinal anesthesia with the patient in prone position.

## Percutaneous repair

All acute injuries with a defect size smaller than 3 cm were subjected to percutaneous repair (PR). PR was performed by making a total of eight incisions, each measuring 1 cm, on the medial and lateral sides of the Achilles tendon. A reinforced suture was used to pass from proximal to distal (at a 45° angle) using a single needle, and the suture ends were tied together on the distal medial side (Fig. 3).

## Primary repair and FHL-A

When the gap was less than 3 cm, delayed ATR was repaired using the Krakow suture technique in an end-to-end. Subsequently, using the same incision, the FHL tendon was released distally with the ankle in maximum plantar flexion. The freed tendon was then

**Table 2 Postoperative casting and rehabilitation program.**

| Rehabilitation | PR | Primary repair+FHL-A | Y plasty and FHL-A |
|---|---|---|---|
| CASTİNG AND CAM | | | |
| Short leg circular cast with 15 degrees plantar flexion | 0–15th day | 0–15th day | 0–15th day |
| Short leg circular cast in neutral position | 2th–4th weeks | 2th–6th weeks | 2th–6th weeks |
| Switching to CAM and starting ROM exercises | 4th–8th week | 6th–10th week | 6th–10th week |
| WEIGHTING | | | |
| Slight weight-bearing on tiptoes | From the 1st day | From the 1st day | From the 1st day |
| As much weight as tolerable | 4th–8th week | 6th–10th week | 6th–10th week |
| Full weighting | From the 8th week | From the 10th week | From the 10th week |
| RETURN TO SPORT AND WORK | | | |
| Sport | From the 6th month | From the 6th month | From the 6th month |
| Work | From the 10th week | From the 12th week | From the 12th week |

fixed using a bioRCI screw and button in the position closest to the Achilles insertion site. Finally, the remaining fibres of the FHL were sutured to the Achilles tendon (Fig. 3).

## V-Y plasty and FHL-A

V-Y plasty was employed for deformities exceeding 3 cm in size. A V-shaped incision, with the arms of the V 1.5 times larger than the defect, was created on the gastrocnemius aponeurosis. The V-shaped section was then carefully elongated until the defect was closed. The tendon ends were brought together using the Krakow procedure, while ensuring that the ankle was in the correct posture. Subsequently, FHL-A was performed. Finally, the paratenon and subcutaneous tissues were appropriately sutured together (Fig. 3).

## Statistical analysis

The normal distribution of the data was assessed using histograms, Q-Q plots, and the Shapiro-Wilk test. Variance homogeneity was tested with the Levene test. For binary group comparisons, the independent two-sample t-test was applied for quantitative variables. For comparisons between more than two groups, one-way analysis of variance was used for quantitative variables. Tukey and Tamhane tests were used for multiple comparisons. Categorical data were analysed using Pearson's chi-squared test. The statistical analysis was conducted using the R 4.3.1 software (*R Core Team, 2023*). The significance level was set at $p < 0.05$.

## RESULTS

The mean forward jump was 142.69 ± 7.14 cm in individuals who received PR, 137.71 ± 4.51 cm in those who received primary repair + FHL-A, and 123.88 ± 3.09 cm in those who received V-Y plasty + FHL-A ($p < 0.001$). The decrease in the mean vertical jump distance on the operated side compared to the contralateral extremity was 0.97 ± 0.93, 2.07 ± 0.99,

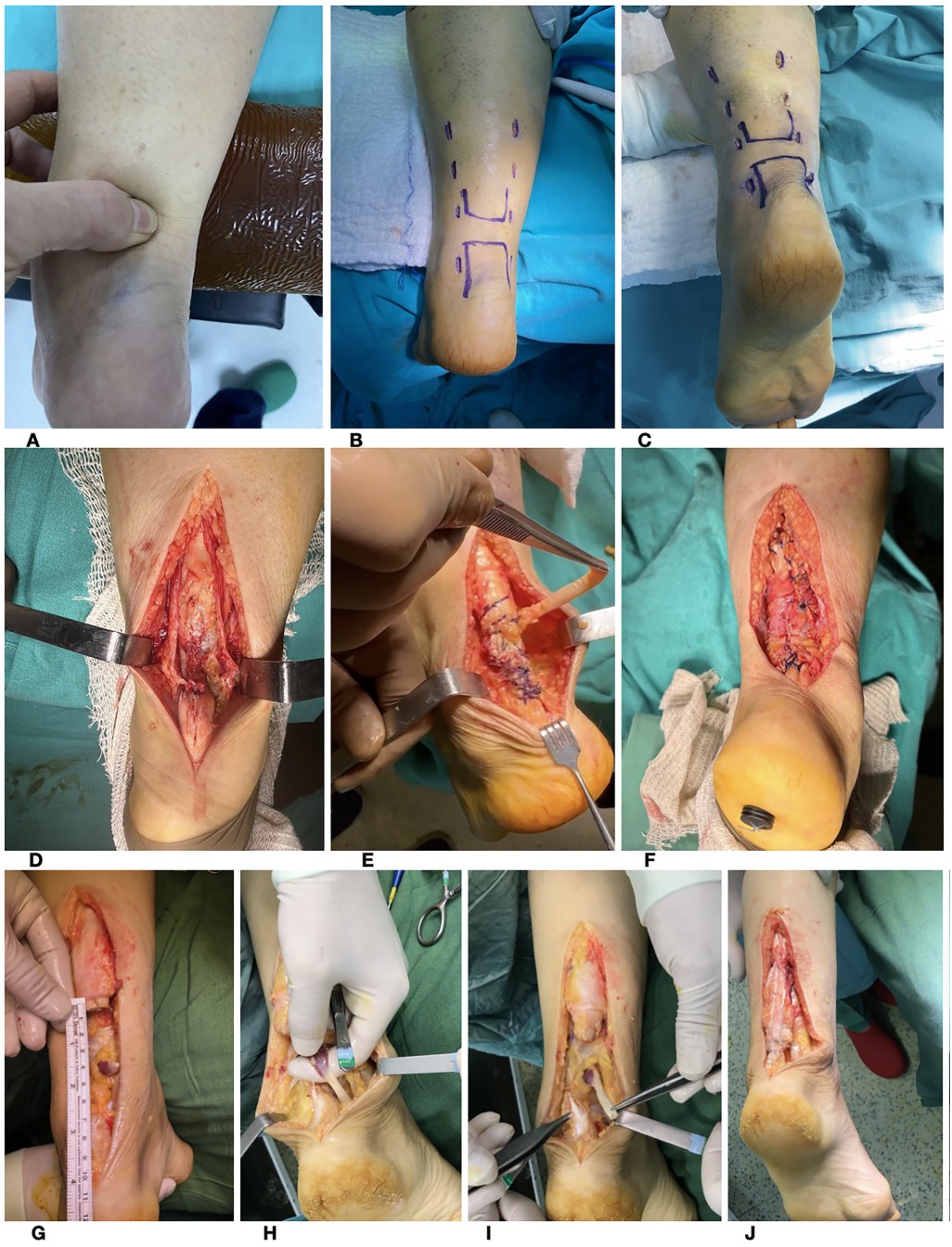

**Figure 3** (A) Palpation of the gap amount. (B) Marking of suture crossing points. (C) View after PR. (D) Exposure of the Achilles tendons for primary repair. (E) View of primary repair and FHL harvest. (F) View after primer repair + FHL-A. (G) Measurement of the gap amount. (H) Preparation of the FHL for transfer. (I) View of FHL harvest. (J) View after V-Y plasty and FHL-A.

and 3.00 ± 1.69 cm in individuals who underwent PR, primary repair + FHL-A, and V-Y plasty + FHL-A, respectively (*p* < 0.001). Table 3 shows the remaining jump performance values.

**Table 3 Functional outcomes comparison based on surgical techniques.**

| VARİABLES | | SURGERY TYPE (*n* = 54) | | | *P* |
|---|---|---|---|---|---|
| | | PR (*n* = 32) | Primary repair+FHL-A (*n* = 14) | V-Y plasty and FHL-A (*n* = 8) | |
| GENDER | | | | | |
| Male | *n* (%) | 24 (75.0) | 12 (85.7) | 6 (75.0) | 0.738 |
| Female | | 8 (25.0) | 2 (14.3) | 2 (25.0) | |
| JUMP PERFORMANCE (centimetre) | | | | | |
| Operated side jump forward jump | mean ± SD | 142.69 ± 7.14[a] | 137.71 ± 4.51[b] | 123.88 ± 3.09[c] | <0.001 |
| Non-operated side forward jump | | 148.06 ± 7.18[a] | 145.29 ± 5.47[a] | 136.00 ± 1.31[b] | <0.001 |
| *p** | | <0.001 | <0.001 | <0.001 | |
| Δ | | 5.38 ± 3.67[a] | 7.57 ± 3.18[a] | 12.13 ± 3.14[b] | <0.001 |
| Operated side vertical jump | | 20.94 ± 1.16[a] | 19.21 ± 1.05[b] | 16.75 ± 0.89[c] | <0.001 |
| Non-operated side vertical jump | | 21.91 ± 0.78[a] | 21.29 ± 0.73[a] | 19.75 ± 1.16[b] | <0.001 |
| *p** | | <0.001 | <0.001 | 0.002 | |
| Δ | | 0.97 ± 0.93[a] | 2.07 ± 0.99[b] | 3.00 ± 1.69[b] | <0.001 |
| ROM (degree) | | | | | |
| Operated side plantar flexion | mean ± SD | 42.44 ± 1.74 | 43.36 ± 1.98 | 42.00 ± 4.31 | 0.344 |
| Non-operated side plantar flexion | | 45.47 ± 1.83[a] | 44.21 ± 2.15[b] | 43.00 ± 4.21[b] | 0.025 |
| *p** | | <0.001 | <0.001 | 0.007 | |
| Δ | | 3.03 ± 1.47[a] | 0.86 ± 0.66[b] | 1.00 ± 0.76[b] | <0.001 |
| Operated side dorsoflexion | | 18.50 ± 1.19[a] | 22.71 ± 1.54[b] | 22.63 ± 0.92[b] | <0.001 |
| Non-operated side dorsoflexion | | 22.84 ± 1.48[a] | 24.64 ± 1.34[b] | 25.00 ± 0.00[b] | <0.001 |
| *p** | | <0.001 | <0.001 | <0.001 | |
| Δ | | 4.34 ± 1.18[a] | 1.93 ± 1.07[b] | 2.38 ± 0.92[b] | <0.001 |
| Operated side Heel rise test | | 14.91 ± 1.51[a] | 15.71 ± 1.20[a] | 12.38 ± 1.41[b] | <0.001 |
| Non-operated side Heel rise test | | 17.69 ± 1.47 | 17.79 ± 1.31 | 18.13 ± 0.64 | 0.714 |
| *p** | | <0.001 | <0.001 | <0.001 | |
| Δ | | 2.78 ± 1.62[a] | 2.07 ± 0.47[a] | 5.75 ± 0.89[b] | <0.001 |
| RETURN TO SPORT AND WORK TİME (week) | | | | | |
| Sport | mean ± SD | 31.19 ± 6.10 | 29.36 ± 4.73 | 34.13 ± 6.92 | 0.200 |
| Work | | 13.84 ± 3.10[a] | 15.57 ± 2.24[ab] | 16.50 ± 1.60[c] | 0.024 |
| RERUPTURE | | | | | |
| Present | *n* (%) | 3 (9.4) | 0 (0.0) | 0 (0.0) | 0.419 |
| Absent | | 29 (90.6) | 14 (100.0) | 8 (100.0) | |
| BALANCE DİSORDER | | | | | |
| Present | *n* (%) | 4 (12.5) | 0 (0.0) | 2 (25.0) | 0.157 |
| Absent | | 28 (87.5) | 14 (100.0) | 6 (75.0) | |
| THUMB WEAKNESS | | | | | |
| Present | *n* (%) | 0 (0.0) | 3 (21.4) | 0 (0.0) | 0.016 |
| Absent | | 32 (100.0) | 11 (78.6) | 8 (100.0) | |

| VARİABLES | | SURGERY TYPE (*n* = 54) | | | *P* |
|---|---|---|---|---|---|
| | | PR (*n* = 32) | Primary repair+FHL-A (*n* = 14) | V-Y plasty and FHL-A (*n* = 8) | |
| INFECTION | | | | | |
| Present | *n* (%) | 0 (0.0) | 2 (14.3) | 4 (50.0) | 0.001 |
| Absent | | 32 (100.0) | 12 (85.7) | 4 (50.0) | |
| NERVE INJURY | | | | | |
| Present | *n* (%) | 4 (12.5) | 1 (7.1) | 0 (0.0) | 0.566 |
| Absent | | 28 (87.5) | 13 (92.9) | 8 (100.0) | |

**Table 3** (*continued*)

**Note:**
Data are expressed as mean ± standard deviation (SD) and number (*n*) (%). The presence of identical letters in the same row indicates the similarity between the surgical types, while distinct letters indicate the difference. Δ: (Nonoperated-Operated).

The decrease in the mean dorsiflexion of the operated side ankle compared to the contralateral extremity was found to be 4.34 ± 1.18, 1.93 ± 1.07, and 2.38 ± 0.92 degrees in individuals who were underwent PR, primary repair + FHL-A and V-Y plasty+FHL-A, respectively ($p < 0.001$) (Fig. 4), respectively. The other ankle ROM measurements are shown in Table 3.

The mean AOFAS score was 88.09 ± 4.34 in individuals who underwent PR, 89.00 ± 4.42 in those who underwent primary repair + FHL-A, and 78.13 ± 5.82 in those who underwent V-Y plasty + FHL-A ($p < 0.001$). The mean VISA-A score was 84.47 ± 5.79 in individuals who underwent PR, 88.14 ± 4.72 in those who underwent primary repair + FHL-A, and 73.88 ± 6.85 in those who underwent V-Y plasty + FHL-A ($p < 0.001$) (Table 4). In individuals who underwent PR, there was no evidence of infection. In contrast, infection was observed in two (14.3%) patients in the group who received primary repair + FHL-A and in four (50%) patients in the group who received V-Y plasty + FHL-A ($p < 0.001$). Four patients (12.5%) who underwent PR experienced nerve injury, while one patient (7.1%) who underwent primary repair with FHL-A repair experienced nerve injury. However, no patients who underwent V-Y plasty with FHL-A repair experienced nerve injury ($p < 0.566$). The number of reruptures and number of individuals with balance disorders, thumb weakness, nerve injury, and other complications between the groups are shown in Table 3.

## DISCUSSION

ATR is a serious injury for which the most effective treatment remains a subject of debate. This study found that a reduction in dorsiflexion following PR resulted in a prolonged recovery time for athletes to return to sports, in comparison to primary repair + FHL-A. Our results demonstrated that, contrary to popular belief, FHL-A does not decrease plantar flexion strength, but rather preserves it and provides a more balanced ankle in chronic rupture.

A study conducted by *Hsu et al. (2015)* demonstrated that open surgical repair yielded favourable clinical outcomes. However, it also resulted in superficial and deep wound
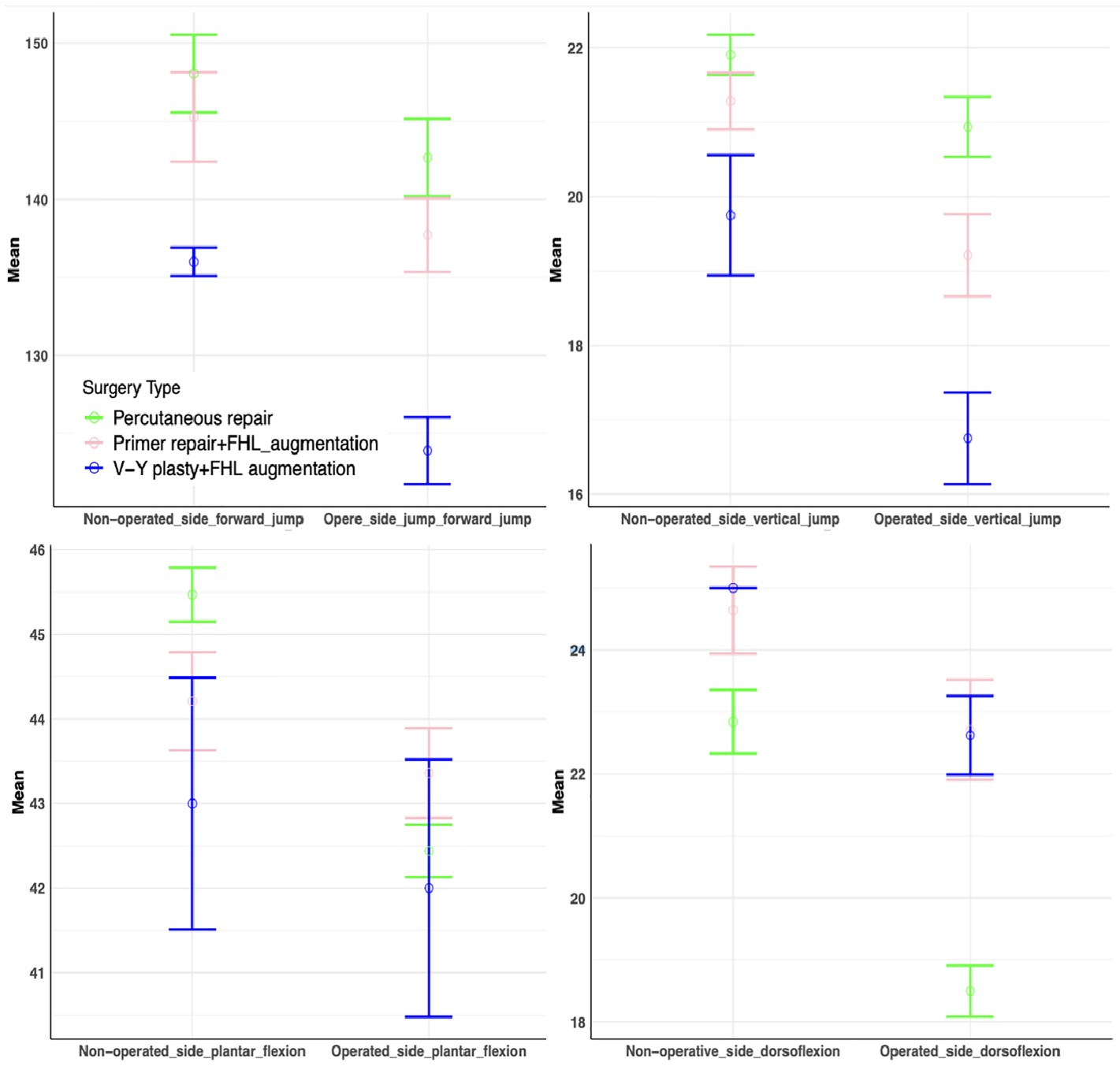

**Figure 4 Error bar of the mean ROM and jump performance values after surgical techniques.**

complications. The most significant complication of open repair is infection. Infections and wound site complications occur with a frequency of 12.5% in the postoperative period (*Kadakia, Dekker & Ho, 2017*). However, larger scars form after open surgical repair, and complications such as necrosis of the surrounding skin and exposure of the Achilles tendon can occur (*Bottagisio & Lovati, 2017*; *Yamamoto et al., 2017*). In this study, infection developed in six patients (36.4%) who underwent open repair ($p < 0.001$). Two of

**Table 4 AOFAS scoring system at 1 year, VISA-A scale at 6 months.** AOFAS, American Orthopedic Foot and Ankle Society; VISA-A, Victoria Institute of Sport Assessment-Achilles.

| AOFAS: American Orthopedic Foot and Ankle Society | | PR: Percutaneous repair | Primary repair+FHL-A: Flexor hallucis longus augmentation | V-Y plasty and FHL-A: Flexor hallucis longus augmentation | |
|---|---|---|---|---|---|
| Score | mean ± SD | 88.09 ± 4.34[a] | 89.00 ± 4.42[a] | 78.13 ± 5.82[b] | <0.001 |
| Excellent | n (%) | 19 (59.4) | 9 (64.3) | 0 (0.0) | |
| Good | | 12 (37.5) | 5 (35.7) | 4 (50.0) | |
| Fair | | 1 (3.1) | 0 (0.0) | 3 (37.5) | |
| Poor | | 0 (0.0) | 0 (0.0) | 1 (12.5) | |
| VISA-A | | | | | |
| Score | mean ± SD | 84.47 ± 5.79[a] | 88.14 ± 4.72[a] | 73.88 ± 6.85[b] | <0.001 |
| Excellent | n (%) | 8 (25.0) | 10 (71.4) | 0 (0.0) | |
| Good | | 18 (56.3) | 3 (21.4) | 3 (37.5) | |
| Fair | | 6 (18.8) | 1 (7.1) | 4 (50.0) | |
| Poor | | 0 (0.0) | 0 (0.0) | 1 (12.5) | |

**Note:**
Data are expressed as mean ± standard deviation (SD) and number (n) (%). The presence of identical letters in the same row indicates the similarity between the surgical types, while distinct letters indicate the difference.

these infections (14.3%) were observed in open surgical repair and FHL-A patients, and four (50%) in V-Y plasty and FHL-A patients ($p < 0.001$). Five of them had superficial infections that regressed with daily wound care and oral antibiotic therapy. However, one patient developed a foreign body reaction due to the non-absorbable braided suture material, resulting in a deep infection. This patient demonstrated healing with the debridement of the infected tissues and application of antibiotic therapy, which negatively affected their clinical results. The most probable reasons for the high incidence of infection in patients who performed V-Y plasty in our study were prolonged plantar flexion, use of non-absorbable braided suture material, and application of the technique with long incisions. High infection rates and delayed surgical treatment led to further loss of muscle strength. The combination of V-Y plasty and FHL-A resulted in a more pronounced decline in jump performance, heel rise tests, and AOFAS and VISA scores compared to the uninjured extremity in individuals with V-Y plasty and FHL-A in the other two procedures.

It has been reported that the combination of early weight-bearing and ankle movement exercises is more beneficial for postoperative recovery compared to either typical immobilization or early ankle movement exercises alone (*Huang et al., 2015*; *Tarantino et al., 2020b*). Moreover, research has demonstrated that the rates of rerupture vary between 6% and 34% following different rehabilitation programmes or intensive workouts (*Braunstein et al., 2018*; *McCormack & Bovard, 2015*). In the study, patients were permitted to bear partial weight on their tiptoes starting from the first day following surgical intervention. By the end of the fourth week, patients were allowed as much weight as tolerated and also began engaging in passive ankle motions. The majority of patients

who underwent PR were active athletes, and the physiotherapy protocol was implemented 15 days prior to those who received open surgical repair. Patients treated with PR demonstrated better jump performance compared to those treated with the other two approaches. For the evaluation, the mean decrease in forward jump performance was 5.38 ± 3.67 units in treated individuals compared to the uninjured extremity, while it was 7.57 ± 3.18 and 12.13 ± 3.14 in the open surgical repair + FHL-A and V-Y plasty + FHL-A groups, respectively ($p < 0.001$). Higher jumping performances were correlated with AOFAS scores, which averaged 88.09 ± 4.34 in individuals who underwent PR. However, PR resulted in higher rates of re-rupture. Although PR technique improved clinical outcomes with better postoperative blood supply and accelerated tissue healing, we think that rerupture rates were higher in this technique because fewer tendon fibers were captured and attached end to end. Moreover, earlier rehabilitation programs have been demonstrated to enhance performance tests, although this can potentially increase the risk of rerupture.

Although a definitive assessment method has not yet been found, the self-administered VISA-A questionnaire is currently the most widely-accepted method used to measure the severity of after-ATR repair (*Becher et al., 2018*; *De Carli et al., 2016*) or Achilles tendinopathy by assessing the impact on pain, function, and activity (*Robinson et al., 2001*). In this study, a mean decrease of 4.34 ± 1.18 in the degree of dorsiflexion was observed in the individuals who underwent PR compared to the uninjured extremity after the operation, while these values were 1.93 ± 1.07 and 2.38 ± 0.92 in the open surgical repair +FHL-A and V-Y plasty + FHL-A groups, respectively ($p < 0.001$). Previous studies reported that athletes who underwent PR due to ATR achieved satisfactory results in returning to their pre-injury levels (*Tarantino et al., 2020a*; *Ververidis et al., 2016*). However, in this study, the reduction in dorsiflexion was attributed to painful ankles observed in individuals who received PR. Similarly, *Hockenbury & Johns (1990)* reported greater loss of dorsiflexion strength after PR compared with open repair. This condition also resulted in a prolonged return to sport. The discrepancy between the VISA-A and AOFAS scores in individuals who underwent PR in the study can be attributed to the painful Achilles healing observed in this population.

A frequently observed consequence of PR is the development of sural nerve injury. Studies demonstrate that sural nerve injury may occur in 0–27% of patients. To minimize the chances of complications, a combination of percutaneous and open techniques may be utilized (*Jiang et al., 2021*; *Kakiuchi, 1995*). Sural nerve injury was observed in four (12.5%) patients who underwent PR and in one (7.1%) patient in the open surgical repair + FHL-A group. In the four patients who underwent PR, the sural nerve damage healed completely within a period of 5 months. However, in one patient who underwent open surgical repair + FHL-A, problems continued to persist for an extended duration. The patient experienced electrified sensation and plantar paresthesia, accompanied by supination of the ankle. By the conclusion of the second month, the electrified sensation had disappeared, but the plantar paresthesia persisted. Modulation of inflammation in the early stages following tendon repair improves repair (*Hays et al., 2008*). Regulated inflammation is largely beneficial for tendon repair, but excessive or prolonged inflammation can be detrimental

and lead to poor clinical outcomes (*Sugg et al., 2014*; *Lichtnekert et al., 2013*). We could say that the electrical sensation experienced in this study was caused by the uncontrolled proliferation of inflammatory tissue after tendon repair, due to tension on the sural nerve or neighbouring tissues around it.

FHL-A transfer is a recommended technique for the treatment of ATR associated with large tendon defects and neglected Achilles rupture. When performed with a single incision, this technique has minimal morbidity and complications and can provide excellent functional and clinical results along with patient satisfaction (*Abubeih et al., 2018*; *Yassin et al., 2021*). However, *Suttinark & Suebpongsiri (2009)* reported a limitation in dorsiflexion following FHL transfer. Prior research demonstrated that FHL-A did not yield any additional benefits and, in fact, led to an increase in the formation of scar tissue, thereby adversely impacting functional outcomes (*Wapner et al., 1993*). However, in this study, decreases in the degrees of plantar flexion and dorsiflexion were fewer in those who performed FHL-A compared to the contralateral extremity. This difference was statistically significant ($p < 0.001$). In the balance test, the FHL-A treated groups demonstrated superior outcomes. The FHL-A made a substantial contribution to maintaining ankle ROM and balance in individuals with chronic ruptures. This may be explained by the fact that the FHL regains size to the retracted Achilles tendon and extends in the similar direction with the Achilles tendon, forming the ankle plantar muscle group together (*Abubeih et al., 2018*; *Gerstner et al., 2021*; *Yeoman, Brown & Pillai, 2012*). Furthermore, it is also proposed that FHL-A has critical importance for individuals with ruptures in the vicinity of the calcaneal insertion site (greater than 2 cm), as healing is challenging (*Campillo-Recio et al., 2021*).

It is well established that weakness in the plantar flexion of the big toe is a known morbidity of FHL-A. The clinical significance of this morbidity is considered to be negligible and does not lead to functional disability (*Wapner et al., 1993*; *Coull, Flavin & Stephens, 2003*). In this study, three patients who underwent FHL-A exhibited weakened big toe plantar flexion, while the remainder did not (Table 3). In particular, in sports such as soccer where lower extremity strength is of paramount importance, this complication should be considered when considering FHL-A.

## Limitations and strengths

The evident limitations that undermined our observations were the limited patient sample and retrospective design of our study. Consequently, prospective cohort studies and randomized controlled trials are necessary to enhance the data base that informs surgeons in the management of patients with ATR. Additionally, the use of different suture materials, not all injuries being evaluated as acute or chronic, and the groups not being equally distributed were disadvantages of this study.

Our findings were obtained from a carefully chosen and limited cohort of individuals who underwent surgery at two different specialized institutions due to ATR. This may have enhanced the internal validity of the study but restricted the generalizability of the findings. Subsequent research should endeavor to incorporate a more heterogeneous patient demographic across several institutions to validate the results.

An important aspect of the current study is that it includes performance tests involving the intact extremity, which allows for a more comprehensive understanding of the findings. Additionally, this is one of the rare studies that incorporates the performance decline of the injured extremity into intergroup comparisons. Another important aspect of the study is its contribution to the improvement of performance tests in athletes after Achilles rupture, its focus on which surgical method might be more effective, and its emphasis on the quality of general sports activities post-injury.

## CONCLUSION

Although no surgical technique is completely superior to another, better performance tests were observed after PR repair compared to open surgery in individuals involved in sports, but rerupture, dorsiflexion restriction, and painful ankle are disadvantages. FHL-A, which has gained popularity in recent years, can improve performance tests by contributing to a more stable and stronger ankle in in sporting individuals with chronic ruptures who have undergone open surgery. FHL-A can be utilized in addition to the primary surgical intervention in individuals with high performance expectations before and after injury.

### Funding
The authors received no funding for this work.

### Competing Interests
The authors declare that they have no competing interests.

### Author Contributions
- Hüseyin Kürüm conceived and designed the experiments, prepared figures and/or tables, and approved the final draft.
- Hacı Bayram Tosun conceived and designed the experiments, authored or reviewed drafts of the article, and approved the final draft.
- Faruk Aydemir performed the experiments, authored or reviewed drafts of the article, and approved the final draft.
- Orhan Ayas conceived and designed the experiments, performed the experiments, prepared figures and/or tables, and approved the final draft.
- Kübra Orhan Kürüm analyzed the data, authored or reviewed drafts of the article, and approved the final draft.
- Funda İpekten analyzed the data, prepared figures and/or tables, and approved the final draft.

### Human Ethics
The following information was supplied relating to ethical approvals (*i.e.*, approving body and any reference numbers):

Fırat University Hospital Human Subject Research Ethics Committee (2023/13–38).

## Data Availability

The raw measurements are available in the Supplemental File.

## Supplemental Information

Supplemental information for this article can be found online at http://dx.doi.org/10.7717/peerj.18890#supplemental-information.

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
