# Peer review of "Surgical outcomes in patients with Achilles tendon rupture—a retrospective study"

_PeerJ, doi:10.7717/peerj.18890_

## Round 0.1 · original submission · Major Revisions

The authors are requested to carefully revise the manuscript and answer the questions raised by the reviewers.

Reviewer 1 ·

Basic reporting

no comment

Experimental design

no comment

Validity of the findings

no comment

Additional comments

1.As the authors pointed out, there are many methods for treating both acute and chronic ATR, many of them still remain controversial in orthopedic surgery. why this article chosen these three surgical methods for study was not cited in detail. Thus, the introduction section needs to be supplemented with more detailed information to illustrate the significance of this study. I suggest that you improve the description at lines 53-62 to provide more justification for your study (specifically, you should expand upon the knowledge gap being filled to fully explain the need and importance of this study).
2.In the “surgical technique” section, the study only provided the figure of V-Y plasty+FHL-A, the figures of PR and primary repair+FHL-A techniques were not provided. Although both of them are common methods in the treatment of ATR, it is best to provide pictures for a more visual illustration. Additionally, the figure 2 provided in the paper was not sufficiently clear and needed to be enhanced and added markings (such as FHL, broken sites after repaired) and illustrations for publication.
3.In the “rehabilitation” section, there are many rehabilitation procedures in the three surgical techniques. Is it the author's experience or a reference to other authors' experience? If it is the author's experience, the description is currently confusing, a tabular description is preferable. If it is the experiences of other authors, pay attention to noting the citations.

1. Hsu AR, Jones CP, Cohen BE, Davis WH, Ellington JK, Anderson RB. Clinical outcomes and complications of percutaneous Achilles repair system versus open technique for acute Achilles tendon ruptures. Foot Ankle Int. 2015;36:1279.1286.
2. Jiang, X., Qian, S., Chen, C., Wu, H., Zhi, X., Xu, D., Lian, J., Liu, X., Wei, S., & Xu, F. Modified mini-incision "internal splinting" versus percutaneous repair technique of acute Achilles tendon rupture: five year retrospective case-controlled study. International orthopaedics 2021;45:3243.3251.
3. Abubeih H, Khaled M, Saleh WR, Said GZ. Flexor hal- lucis longus transfer clinical outcome through a single incision for chronic Achilles tendon rupture. Int Orthop. 2018;42:2699.2704.

Reviewer 2 ·

Basic reporting

The use of English language need to be improved. Some words need to be replaced with more appropriate ones (e.g., line 64, "surgery" o "surgical intervention" instead of "operations" or line 86, "ascending an incline", line 127, line 267 "electrification", etc), acronyms checked, as well as typing mistakes.
Please ask for a review of the entire manuscript by a native English speaker.

Referece list is generally fine even if short and with some quite old articles. I suggest to improve them incorporating the following articles:

- Tarantino, D., Palermi, S., Sirico, F., Balato, G., D’Addona, A., & Corrado, B. (2020). Achilles tendon pathologies: How to choose the best treatment. Journal of Human Sport and Exercise, 15(4proc), S1300-S1321. Retrieved from https://www.jhse.ua.es/article/view/2020-v15-n4-proc-achilles-tendon-pathologies-best-treatment

- Maffulli, Nicola; Aicale, Rocco; Tarantino, Domiziano. Autograft Reconstruction for Chronic Achilles Tendon Disorders. Techniques in Foot & Ankle Surgery 16(3):p 117-123, September 2017. | DOI: 10.1097/BTF.0000000000000154

- Tarantino D, Palermi S, Sirico F, Corrado B. Achilles Tendon Rupture: Mechanisms of Injury, Principles of Rehabilitation and Return to Play. J Funct Morphol Kinesiol. 2020 Dec 17;5(4):95. doi: 10.3390/jfmk5040095. PMID: 33467310; PMCID: PMC7804867.

Figures and tables are good but some of them should be moved to other sections of the manuscript.

Experimental design

The research question is not well defined, since it is not clear the nature of the study as well as its scopes.

Patients' randomization is not clear, inclusion criteria were not reported, and the exclusion ones are very poorly reported.

Methods are not sufficiently described and the way they were reported is quite confusing.

Validity of the findings

The quality of the findings is good: however, a strong comparison with the available scientific literature was not performed, thus limiting the novelty of the manuscript to a mere report of the study's results.

Furthermore, apart from the retrieved outcomes, some reported assumptions lack of scientfic references.

Conclusions are very poor.

Additional comments

INTRODUCTION

Line 44: please explain acronyms when reporting them for the first time in the manuscript. The same at line 47.

Lines 48-49: you should better state that "chronic" rupture are those that do not recover after 6 months of conservative treatment.

Lines 57-62: was your aim to just report the outcomes of the different techniques or to compare them? This information should be clearly reported.

Brief information about PR, FHL and V-Y plasty for ATR should be reported in this section to give an insight into these surgical techniques.

METHODS
Line 60: since the VISA-A is mainly used for Achilles tendinopathy, why did you not use the ATRS which is more specific for ATR?

Lines 65-66: how were patients rendomized to one of the three surgery groups? Even if this information can be retrieved further on, a brief explanation about it should be reported here.

Line 65: evaluated as "acute" and "chronic" basing on which criteria? Please explain how they were assessed.

Lines 72-77: no information about patients' demographics and/or health issues were reported.

Forward and Vertical Jump Performance and Single Leg Stance test: pictures of both tests should be reported.

Line 97: their srength? The strength of what?

Lines 119-120: rephrase. Did you mean after surgery?

RESULTS
Lines 168-171: these information are redundant. Delete.

Table 1 shoud be moved in the Methods section

Lines 146-147: was FHL augmentation or V-Y plasty performed in already operated patients? This because, at line 80, you stated that you excluded patients "who had undergone revision Achilles surgery", while here it seems that those surgeries were performed in patients where PR alone was unsuccessful.

DISCUSSION
Line 224-226: since "...early weight-bearing and early ankle movement exercises is more beneficial for postoperative recovery than either typical immobilization", why did you prefer to immobilize the operated leg? Furthermore, why did you prefer the neutral position instead of the equinus one?

Lines 241-242: it is not correct to assess that "earlier rehabilitation programmes...facilitate the occurrence of rerupture". You may say that, given the early rehabilitation, the risk of re-rupture can be increased, but not that early rehabilitation facilitate re-rupture by default.

Lines 243-246: why did you report on Achilles tendinopathy if you performed a study on ATR? You may state, in the Introduction, that chronic Achilles tendinopathy is a risk factor for ATR, but no further information are reported since they seem out of context.

Line 248-249: VISA-A is reliable for Achilles tendinopathy: do you have any reference that it could be also used for ATR?

Lines 256-257: redundant information. Merge with the previous reported ones.

Lines 269-270: how could be the "increased quantity of healing tissue" responsible for the persistence of plantar paresthesia?

Lines 282-284: "...that the FHL regains size to the retracted achilles tendon and extends in the similar direction with the Achilles tendon, forming the ankle plantar muscle group together". How did you suppose that? Report any evidence from the scientific literature. Same at lines 285-286.

Lines 294-296: these lines could not be considered as "Strengths and limitations". Those lines need to be substantially improved.

CONCLUSIONS
Line 299: where was tendinopathy reported as a "disadvantage" of PR?

Lines 303-305: copied and pasted from lines 294-296. Thiese could not be accepted as proper conclusions for a scientific study.

---

## Round 0.2 · Minor Revisions

The authors are requested to carefully revise the manuscript and answer the questions raised by the reviewers.

Reviewer 1 ·

Basic reporting

no comment

Experimental design

no comment

Validity of the findings

no comment

Reviewer 2 ·

Basic reporting

Check for terms "tendinopathy" and "tendinitis" and please only use the first one.

The use of academic English has been improved but there are still some major flaws (line 165 "6rd", poorly written sentences and several typing mistakes) throughout the entire manuscript that need to be checked and fixed. Please double check the manuscript before resubmission.

Experimental design

When talking about rehab, you did not report in which position were patients who underwent open surgery immobilized. Please add this information.

Validity of the findings

Comment and answer nr. 22 in the attached file about the reported information on Achilles tendinopathy: you probably misunderstood what I asked you.

I was skeptical about the need of reporting information on Achilles tendinopathy in the Discussion (lines 295-301 in the new version of the manuscript) since they are out of context. Furthermore, right after, you reported about the outcomes of the surgical techniques.
Even your answer is not clear (what did you mean with "This situation led to low VISA-A scores in individuals who underwent percutaneous repair. Achilles tendonitis has been shown as a cause for ATR in the literatüre (1-3). However, in our study, it was observed as a result of PR. That is why we mentioned this situation similar to Achilles tendonitis."?).

Further on, at comment and answer 28, you stated that "To our knowledge, this study is the first to report tendinopathy following PR". Did you clearly report this information in the Results and/or Discussion? Did you give a possible explanation about it?

Strengths and limitation are still scarce and poorly written. Please rephrase and improved them.

---

## Round 0.3 · Minor Revisions

The authors are requested to carefully revise the manuscript in response to the questions raised by the reviewer. We will be unable to move your article forward unless the language is improved.

Reviewer 2 ·

Basic reporting

There is still heterogeneity in the use of terms such as "tendinopathy", "tendinitis", etc. I clearly asked you to just use "tendinopathy".

The Authors still have difficulties with the use of academic English, so I will leave the decision on the acceptability of the manuscript to the Editor because you probably did not fully understand my requests.

Experimental design

No further comments

Validity of the findings

No further comments

Additional comments

No further comments

---

## Round 0.4 · accepted · Accept

After revisions, all reviewers agreed to publish the manuscript. I also reviewed the manuscript and found no obvious risks to publication. Therefore, I also approved the publication of this manuscript.